# Family Counseling after the Diagnosis of Congenital Heart Disease in the Fetus: Scoping Review

**DOI:** 10.3390/healthcare11212826

**Published:** 2023-10-26

**Authors:** Sophia Livas de Morais Almeida, Luisa Tiemi Souza Tuda, Marcela Bezerra Dias, Luana Izabela Azevedo de Carvalho, Thayla Lais Lima Estevam, Ana Luiza Menezes Teles Novelleto, Edward Araujo Júnior, Luciane Alves da Rocha Amorim

**Affiliations:** 1Postgraduate Program in Health Sciences, Medical School, Federal University of Amazonas (UFAM), Manaus 69067-005, AM, Brazil; 2Medical School, Federal University of Amazonas (UFAM), Manaus 69067-005, AM, Brazil; 3Psychology Sector, Regional Hospital of Taubaté, Taubaté 12030-180, SP, Brazil; 4Medical School, Amazonas State University (UEA), Manaus 69850-000, AM, Brazil; luana.izabela@ebserh.gov.br; 5Department of Obstetrics, Paulista School of Medicine, Federal University of São Paulo (EPM-UNIFESP), São Paulo 04023-062, SP, Brazil; 6Medical School, Municipal University of São Caetano do Sul (USCS), São Caetano do Sul 09521-160, SP, Brazil

**Keywords:** congenital heart disease, fetal heart, family counseling

## Abstract

Congenital heart disease (CHD) is the leading cause of death from malformations in the first year of life and carries a significant burden to the family when the diagnosis is made in the prenatal period. We recognize the significance of family counseling following a fetal CHD diagnosis. However, we have observed that most research focuses on assessing the emotional state of family members rather than examining the counseling process itself. The objective of this study was to identify and summarize the findings in the literature on family counseling in cases of diagnosis of CHD during pregnancy, demonstrating gaps and suggesting future research on this topic. Eight databases were searched to review the literature on family counseling in cases of CHD diagnosis during pregnancy. A systematic search was conducted from September to October 2022. The descriptors were “congenital heart disease”, “fetal heart”, and “family counseling”. The inclusion criteria were studies on counseling family members who received a diagnosis of CHD in the fetus (family counseling was defined as any health professional who advises mothers and fathers on the diagnosis of CHD during the gestational period), how the news is expressed to family members (including an explanation of CHD and questions about management and prognosis), empirical and qualitative studies, quantitative studies, no publication deadline, and any language. Out of the initial search of 3719 reports, 21 articles were included. Most were cross-sectional (11) and qualitative (9) studies, and all were from developed countries. The findings in the literature address the difficulties in effectively conducting family counseling, the strengths of family counseling to be effective, opportunities to generate effective counseling, and the main challenges in family counseling.

## 1. Introduction

Congenital heart disease (CHD) is the most common cause of congenital malformation in the fetus [1,2] and the leading cause of death from malformations in the first year of life [3,4]. Fetal cardiology has been evolving in recent years thanks to technological advances and more significant learning by professionals involved in this subject. Consequently, there was an increase in the diagnosis of intrauterine CHD [1,2,3,4,5].

It has become more evident that when CHD is diagnosed in a fetus, parents need to be informed in a compassionate manner. This requires good communication skills and an instructive explanation of the normal heart’s structures and the affected heart. This approach has been noted to be more effective in helping parents understand the situation [6,7]. Questions about the impairment of the well-being of the fetus during pregnancy, during labor, and after the birth, the need for surgery, what type of surgery, the timing of surgical correction, life quality, and prognosis of the baby are common in any circumstance in the face of some cardiac alteration detected in the intrauterine period. Talking to parents about CHD in the fetus raises many doubts, and counseling is fundamental for families. We recognize the significance of family counseling following a fetal CHD diagnosis. However, we have observed that most research focuses on assessing the emotional state of family members rather than examining the counseling process itself. Therefore, the objective of this study was to identify and summarize the findings in the literature on family counseling in cases of diagnosis of CHD during pregnancy, demonstrating gaps and suggesting future research.

## 2. Materials and Methods

As family counseling in cases of CHD diagnosed during pregnancy is a highly relevant and still emerging topic in the literature, the method chosen was the synthesis of the scope review based on the principles reported by Arksey and O’Malley [8], advocated by the Joanna Briggs Institute principles. We followed the scope review checklist of the Preferred Reporting Items for Systematic reviews and Meta-Analyses extension for Scoping Reviews (PRISMA-ScR) [9].

This methodology included five steps: identification of the research question, identification of relevant studies, selection of studies, data mapping, and demonstration of results. The protocol for this scoping review was registered in the Open Science Framework on 4 September 2022 (https://doi.org/10.17605/OSF.IO/7WK45). As a scoping review, it was not necessary to obtain ethical approval.

### 2.1. Identification of the Research Question

Considering that in the existing literature, the vast majority relates to family counseling after the diagnosis of CHD in children or about the mother’s emotional state, we decided to evaluate the literature regarding the family counseling after the fetal diagnosis of CHD. Therefore, our research question was what do we know from the literature about family counseling after the fetal diagnosis of CHD?

### 2.2. Identification of Relevant Studies

A systematic search identified the studies carried out during September and October 2022. Six databases were used, Medline, Embase, LILACS, Scielo, Scopus, and Web of Science, in addition to the gray literature: PsycINFO and Google Scholar. A health sciences research librarian contributed to the development of the search strategy. The descriptors used were congenital heart disease, fetal heart, and family counseling (Table 1). All studies were securely transferred to the Rayyan system for analysis.

### 2.3. Selection of Studies

The inclusion criteria for the studies were as follows: counseling of family members diagnosed with CHD in the fetus. Family counseling was defined as any health professional who advises mothers and/or fathers regarding the diagnosis of CHD during pregnancy. The studies should cover how to convey the diagnosis of CHD to family members, including an explanation of CHD, and management and prognosis questions. The studies could be empirical or qualitative, quantitative, with no publication deadline, except for Google Scholar, and in any language.

We excluded articles with incomplete text, book chapters, abstracts from congress annals, editorials, and lectures.

The first screening performed was the exclusion of duplicate articles. Next, we had two reviewers working as a pair, reading the titles and abstracts of all publications, following the eligibility criteria. After this step, the full text was read, thus concluding the last phase of screening of studies. Doubts and disagreements were resolved by consensus and discussion. A third reviewer read the full text when it was necessary. We conducted a thorough search to cross-check articles related to the subject matter.

### 2.4. Data Mapping

A Microsoft Excel spreadsheet was used to create a data extraction form (Microsoft Corp., Redmond, WA, USA). The data extracted included the authors, year of publication, study location, objectives, methodology, counseling team, and essential study results.

We listed the selected articles in a single table, from 1 to 21, to facilitate understanding and identification of the graphs and figures.

### 2.5. Results Demonstration

We presented our findings through narratives, tables, and diagrams. The main topics addressed are shown in graphs. The thematic approach was divided into four themes: (1) obstacles to counseling; (2) strengths for effective counseling; (3) opportunities; and (4) challenges for the healthcare services and the health team.

## 3. Results

We found 3719 articles in the initial search of the eight databases. Out of the total articles, 1165 were duplicated and 2535 were excluded after analyzing their titles and abstracts. After analyzing for cross-referencing, there were only 21 articles left (Figure 1).

Table 2 presents the data of the 21 eligible articles including authors, year of publication, study location, objectives, methodology, counseling team, and essential study results. For ease of identification, they were numbered from 1 to 21, also used in Table 3.

Family counseling in cases of fetal heart disease has been the subject of numerous studies worldwide. The United States of America has produced the largest number of publications on this topic (six articles).

We analyzed 21 articles, of which 19 were published in the last decade. Regarding study design, we found that 11 studies were cross-sectional, 9 used qualitative methods, and only one was a randomized clinical trial, as shown in Table 2.

We categorize our results thematically to present them more effectively in the following way: identifying the obstacles that hinder effective family counseling in cases of fetal heart disease diagnosis, highlighting the strengths that contribute to effective family counseling, outlining the opportunities as well as the challenges that services and healthcare teams face with regard to providing appropriate family counseling in such cases (Table 3).

## 4. Discussion

We identified 21 studies focused on counseling families dealing with fetal heart disease during pregnancy. These publications, primarily from the last decade, underscore the contemporary significance of this matter within fetal cardiology centers. Our analysis emphasizes the necessity for additional research in specific global regions and underscores the crucial requirement for effective dissemination of knowledge and improved communication strategies for conveying pertinent information to family members (Table 2). 

We employed a thematic approach to enhance clarity in presenting our findings and organizing the discussion according to the identified themes. The studies included may appear repeatedly, and the main characteristics were analyzed separately (Table 3).

### 4.1. Obstacles to Effective Family Counseling in Cases of Diagnosis Heart Disease in the Fetus 

Effective family counseling can be challenging when a fetus has been diagnosed with CHD. Obstacles often discussed in the 21 articles included the strength of the doctor–patient relationship, the quality of information transmission to the patient, and the standardization of medical terminology.

The obstacles encountered are interconnected, as a strong doctor–patient relationship depends on good communication between the medical team and the family members, requiring the transmission of information using didactic language. The doctor–patient relationship is a crucial aspect of healthcare. It involves two-way communication and trust between the family and their healthcare provider. A strong relationship can result in better health outcomes. To clarify CHD, it is also necessary to standardize the technical terms used, considering that the diseases can be complex and challenging to understand [3,5,6,10,11,12,13,15,16,18,19,20,26].

The studies highlight several flaws in family counseling services. These include a lack of confidence from the health team when it comes to diagnosing, treating, and predicting outcomes, difficulty in explaining the disease clearly and concisely, no private space for family consultations, no follow-up with the family after diagnosis, a prolonged wait time for the diagnosis to be clarified, and a need for a translator in cases where foreign pregnant women are involved [2,3,6,7,10,12,14,15,16,17,18,19,20,24,25].

### 4.2. Strengths for Effective Family Counseling

Strengths identified for effective family counseling in the 21 articles were understanding heart disease, how the diagnosis is communicated, and good infrastructure.

It is crucial for family members to understand the disease and its complexities to comprehend survival, treatment, and prognosis. The use of standardized medical terminology is essential in this regard. The medical team must maintain a consistent, accessible, and understandable language that is comprehensible to all family members. To achieve this, the multidisciplinary team should find ways to make the language intelligible. This approach helps to build trust in the team [5].

Continuous family counseling, a prepared multidisciplinary team, and the presence of a pediatric cardiologist are factors in facilitating appropriate family counseling in cases of CHD with an intrauterine diagnosis [11,25,27]. Some studies have emphasized the importance of translation services and the need for private rooms, audiovisual materials, and adequate time for dialogue in the native language [7,11]. Also, the internet offers a valuable opportunity to enhance communication between physicians and parents by collecting feedback from parents across multiple institutions [6,13,17].

Previous studies indicate that implementing a standardized counseling process improved parents’ comprehension of the medical condition [7,9,10]. Visual aids, such as drawings and written information, can help parents recall essential details during a consultation and inform family decisions about pregnancy treatment options [11]. Then, the presentation of information about CHD can influence parental perceptions and decision making regarding survival rates [13]. 

A good infrastructure is essential for any system to function correctly. Considering our context, we believe that physical and organizational structures and facilities are necessary for effective counseling and the treatment and follow-up of the fetus with CHD [3,14,16,17,23].

### 4.3. Opportunities for Services and the Healthcare Team in Relation to Appropriate Family Counseling

We recognize the importance of providing better family counseling for pregnant women who receive a fetus diagnosis of CHD and the need for a more compassionate and understanding approach that fosters trust between the medical team and the family [2,3,6,7,10,11,12,13,14,15,16,17,18,19,20,21,22,23,24,25,26]. 

This trust empowers the family to make informed decisions about the type of treatment and where their child will receive care. A multidisciplinary support network for family members is crucial in strengthening the doctor–patient relationship and ensuring better access to information about the child’s condition [15,16,17,18,19,20,23,26].

The presence of specialists, such as a sociologist, social worker, nurse, psychologist, palliative care specialist, pediatric and fetal cardiologist, maternal-fetal medicine specialist, geneticist, pediatrician, and, in some cases, a pediatric surgeon, can strengthen the multidisciplinary team. The 21 selected studies mentioned these health professionals, highlighting the importance of a specialized group to counsel these pregnant women. Particular emphasis was given to pediatric and fetal cardiologists for their effectiveness in clarifying fetal heart conditions among specialists [6,15,16,17,18,19,20,23]. 

Humanization and support through a multidisciplinary team can enhance the family’s trust in the medical team and their comprehension of the diagnosis of CHD [3,6,10,11,12,13,14,16,18,19]. As a result, the family becomes more self-assured in making decisions, and this strengthens the bond between parents and children [6,10,13,18,19].

### 4.4. Challenges for Services and the Health Team in Relation to Adequate Family Counseling in These Cases

The most impactful challenges reported in this scoping review were continuous counseling during pregnancy, a pediatric and fetal cardiologist at the time of a suspected diagnosis, a multidisciplinary team prepared to manage situations of CHD, and didactic and reliable audiovisual resources to help explain heart disease. 

Continuing family counseling is crucial, as previously explained. However, it can pose several challenges. One of the significant challenges is the need for a strong bond between the multi-disciplinary team and the family. Another challenge is using complex medical terminologies that may be difficult for family members to comprehend. Additionally, the family members may experience shock and denial after the diagnosis, making it challenging to engage them in counseling. Moreover, the medical team may need help understanding the family’s sociocultural context, further complicating the counseling process [3,13,15,16,18,19,20,21,26].

It was found that having a pediatric and fetal cardiologist present at the time of diagnosis was helpful to improve the understanding of heart disease. However, it can be difficult to reconcile the moment when heart disease is suspected with the presence of a pediatric cardiologist, as it is often suspected during a routine ultrasound examination of pregnant women [2,7,10,12,13,14,17,18,19,20,21,23,25,26]. 

It is not always possible to have an experienced multi-disciplinary team focused on CHD when the diagnosis is made. Usually, larger hospitals are better equipped to handle such cases. However, many medical services diagnose heart disease before the pregnant woman can reach more specialized centers, which means that the family might not always get the necessary support. The challenge is to provide a multidisciplinary group that can offer sufficient support to the family, even if the diagnosis is made in less affordable services [14,15,16,17,18,19,20,23].

Audiovisual resources can help educate families about heart disease. However, not all media information is reliable, and this is the challenge of the multidisciplinary team, since it is hard to identify which information is most appropriate for each type of heart disease [2,6,7,10,12,18,22,23,24,25].

### 4.5. Study Limitations

The quality of the selected studies was not assessed, which is not mandatory for this type of review.

## 5. Conclusions

This scoping review demonstrated that family counseling in cases of fetal diagnosis of CHD is a current topic but not studied adequately. The findings in the literature address the difficulties in effectively conducting family counseling, the strengths of family counseling to be effective, opportunities to generate effective counseling, and the main challenges in family counseling. This scoping review generates essential content for developing a standardized guide covering important points for effective family counseling in cases of CHD in the fetus.

## Figures and Tables

**Figure 1 healthcare-11-02826-f001:**
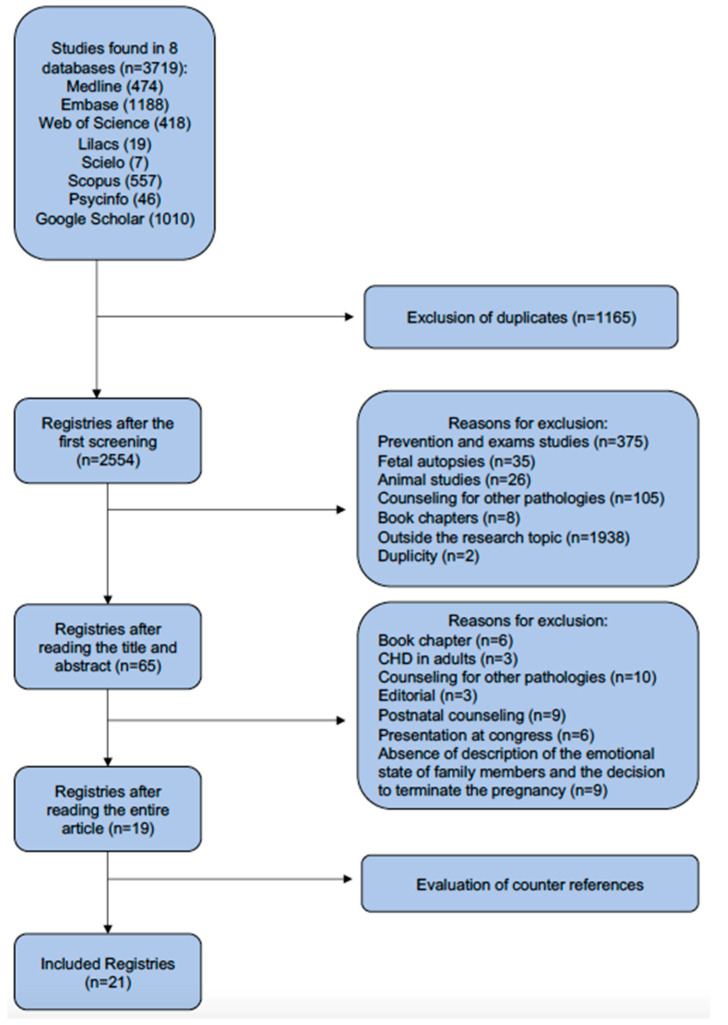
Study’s flowchart.

**Table 1 healthcare-11-02826-t001:** Descriptors used in each database.

Database	Search	Filter	Items Found	Search Date
Medline	((Counseling) AND (“Heart Defects, Congenital” [Mesh] OR “Defect, Congenital Heart” OR “Abnormality, Heart” OR “Heart Abnormality” OR “Congenital Heart Defect” OR “Heart, Malformation Of” OR “Malformation Of Heart” OR “Defects, Congenital Heart” OR “Heart Abnormalities” OR “Heart Defect, Congenital” OR “Congenital Heart Disease” OR “Congenital Heart Diseases” OR “Disease, Congenital Heart” OR “Heart Disease, Congenital” OR “Congenital Heart Defects”) AND (fetal heart))	Humans	474	28 September 2022
Embase	(‘counseling’/exp OR counseling OR ‘parent counseling’/exp OR (counseling, AND parent) OR (guidance, AND parent) OR (parent AND guidance)) AND (‘congenital heart malformation’/exp OR (congenital AND heart AND anomaly) OR (congenital AND heart AND defect) OR (congenital AND heart AND defects) OR (heart AND anomaly) OR (heart AND congenital AND anomaly) OR (heart AND congenital AND defect) OR (heart AND congenital AND malformation) OR (heart AND defects, AND congenital) OR (heart AND malformation)) AND (‘fetus heart’/exp OR (fetal AND heart) OR (foetal AND heart) OR (heart, AND fetus)) AND [Embase]/lim AND [humans]/lim	Humans	1188	28 September 2022
LILACS	(mh:”Aconselhamento” OR (consejo) OR (counseling) OR (conselho) OR mh: f02.784.176* OR mh:f04.408.413* OR mh:n02.421.143.303* OR mh:n02.421.461.363*) AND (mh: “Cardiopatias Congênitas” OR (cardiopatías congénitas) OR (heart defects, congenital) OR (malformação cardiovascular) OR (defeitos cardiovasculares congênitos) OR (anormalidades cardíacas) OR mh:c14.240.400* OR mh:c14.280.400* OR mh:c16.131.240.400*) AND (mh: “Coração Fetal” OR (corazón fetal) OR (fetal heart) OR mh:a07.541.278* OR mh:a16.378.303*) AND (db:(“LILACS”))	Title, abstract and subject	19	28 September 2022
Scielo	((Coração Fetal) OR (Corazón Fetal) OR (Fetal Heart)) AND ((Cardiopatias Congênitas) OR (Cardiopatías Congénitas) OR (Heart Defects, Congenital)) AND ((Aconselhamento) OR (Consejo) OR (Counseling))	No filter	7	1 October 2022
Scopus	(ALL (“Hearts, Fetal” OR “Heart, Fetal” OR “Fetal Hearts”) AND ALL (“Defect, Congenital Heart “ OR “Abnormality, Heart” OR “Abnormality Congenital Heart” OR “Defect Heart, Malformation Of” OR “Malformation Of Heart” OR “Malformation Of Hearts” OR “Defects, Congenital Heart” OR “Heart Abnormalities” OR “Heart Defect, Congenital” OR “Congenital Heart Disease” OR “Congenital Heart Diseases” OR “Disease, Congenital Heart” OR “Heart Disease, Congenital” OR “Congenital Heart Defects”) AND ALL (“counseling” OR “counseling, parent” OR “parent counseling” OR “guidance, parent” OR “parent guidance”))	Medicine; Biochemistry, Genetics and Molecular Biology; Health Professions; Nursing; Multidisciplinary; Psychology	577	1 October 2022
Web of science	fetal heart and congenital heart disease and counseling	No filter	418	1 October 2022
PsycINFO	congenital heart disease AND counseling	No filter	46	1 October 2022
Google Scholar	“fetal heart” and “congenital heart disease” and “counseling”	No filter	first 1010 articles	1 October 2022

**Table 2 healthcare-11-02826-t002:** Main results of the included studies.

Study	Reference	Objective	Methodology	Team	Results
1	Menahem S and Grinwade J. Australia 2004 [10]	Assessment of family members of the counseling provided at the service.	Qualitative study *	PC, FC, and FP **	Counseling improved understanding of CHD and postnatal treatment. Drawings provide a better understanding of CHD.
2	Rempel, GR; et al. Canada. 2004 [11]	Discover and describe how parents make decisions after a diagnosis of CHD during pregnancy and guide professionals in counseling.	Qualitative study	PC, FC **	Healthcare professionals must understand each parent, know their professional influence, and support each family decision.
3	Hilton-Kamm, D; et al. California, USA 2014 [12]	Study the perceptions of parents and experiences upon receiving a diagnosis of CHD.	Cross-sectional study *	PC	The way information is presented to family members can shape parents’ decision making.
4	Carlsson, T; et al. Sweden 2015 [13]	Describe the experiences of parents after a prenatal diagnosis of CHD.	Cross-sectional study *	PC	Valuing early and honest information. The illustrations as a complement to the oral information help in understanding the CHD. Reliable sources on the internet for more information about CHD.
5	Bratt, EL; et al. Sweden 2015 [14]	Describe counseling in cases of fetal CHD and the need for continuous pregnancy monitoring.	Qualitative study *	PC, FC, nurse, obstetric team, psychologist, and social worker **	The short period between suspicion and accurate diagnosis of CHD in the fetus. Continuous counseling during pregnancy. Written and quality information about CHD. Similar explanations for the partner.
6	Carlsson, T et al. Sweden 2016 [15]	Describe the experiences of Swedish immigrant family members diagnosed with CHD in the fetus.	Qualitative study *	Obstetric nurse	The need for an interpreter, visual information, and psychosocial support, and respecting religion regarding the termination of pregnancy.
7	Carlsson, T; et al. Sweden 2016 [6]	Describe the experiences of parents following a prenatal diagnosis of CHD through group discussion.	Qualitative study	PC	Emotional support for the couple is important. Additional information about CHD. Follow-up of the pregnant woman to repeat information about CHD.
8	Walsh, MJ et al. USA 2017 [16]	Assessment counseling for family members with a fetal diagnosis of hypoplastic left heart syndrome.	Cross-sectional study *	PC, FC	Great variability between forms of counseling. Difficulty in addressing the prognosis.
9	Lee, CK. Missouri, USA 2017 [17]	Review the benefits and objectives of counseling in the prenatal stage, the problems encountered, the topics to be covered, and considerations for family support during prenatal care.	Cross-sectional study	PC, FC	Family members believe that cardiologists should increase the amount of information related to CHD from prenatal care and continue for the rest of life.
10	Carlsson, T.; et al. Sweden 2018 [18]	Describe the experiences of parents following a prenatal diagnosis of CHD through group discussion.	Qualitative study	PC	Additional information about CHD. Monitoring the pregnant woman to repeat information about CHD.
11	Kovacevic, A; et al. Germany 2018 [2]	Develop an appropriate questionnaire to assess parental counseling needs.	Cross-sectional study *	PC, FP, and sociologists	The questionnaire applied can be a tool to evaluate the success of family counseling and recommends multidisciplinary counseling.
12	Im, YM; et al. South Korea 2018 [19]	Describe the experience of Korean mothers with prenatal diagnosis of CHD.	Qualitative study	Specialist healthcare professional	Importance of family counseling. The movement of the fetus helps in understanding that the fetus is an independent being and can be influenced by external factors. Importance of the religious and spiritual aspects (Taekyo technique).
13	Bertaud S; et al. England 2020 [20]	Evaluate counseling for family members with a fetal diagnosis of hypoplastic left heart syndrome.	Qualitative study *	PC, FC	Counseling for these family members offers a better view of your child’s prospects.
14	Kovacevic, A; et al. Germany 2020 [21]	Evaluation of family counseling after the diagnosis of CHD in the fetus.	Cross-sectional study *	PC, FP, and sociologists.	Continuous family counseling, with a private room, adequate dialogue time, native language, and use of written or web information to understand CHD. It suggests greater success if performed by a cardiologist.
15	Kovacevic, A; et al. Germany 2021 [3]	Assess the effects of parental counseling for fetal CHD.	Cross-sectional study	PC, FP, and sociologists	Implementing alternative and innovative approaches (e.g., online conferences or virtual reality tools) may aid in facilitating high-quality services in critical times such as in the COVID pandemic.
16	Holmes, KW; et al. Portland, USA 2021 [22]	Assess the understanding of family members of fetal heart defects after counseling.	Cross-sectional study *	PC, FC, and nurse	Fetal counseling was effective in conveying the anatomy and need for surgery. There was less understanding for women with less education. The following were evaluated: description of the cardiac condition, how confident they were in the diagnosis, and whether the fetus would require heart surgery.
17	Kovacevic, A; et al. Germany 2021 [23]	Evaluating family counseling for fetal CHD during COVID-19 pandemic.	Multicenter cross-sectional study *	PC, FC, and PF	There was no significant difference between the groups in relation to the success of counseling these family members even with the COVID-19 pandemic.
18	Delaney, RK; et al. Utah, USA 2021 [24]	To evaluate the effect of two family counseling protocols in cases of CHD in the fetus.	Randomized clinical trial	PC, FC, surgeon, social worker, and palliative care	Publication of the study protocol.
19	Gendler, Y; et al. Israel 2021 [25]	Assessment of family members of the standardized counseling provided at the service.	Cross-sectional study *	PC	Standardized counseling using a checklist helps provide information. Parental satisfaction in the counseling process. Good perception of cardiologists regarding the family understanding of CHD.
20	Kovacevic, A; et al. Germany 2022 [7]	Identify which factors play an important role in the success of family counseling in cases of fetal CHD.	Multicenter cross-sectional study *	PC, FC, and FP	Short time between suspicion and accurate diagnosis of CHD and explanation by the specialist. Information about CHD in a clear and illustrated way.
21	Harris, KW; et al. Pittsburgh, USA 2022 [26]	Describe the experience of parents who receive a diagnosis of fetal CHD (pre and postnatal).	Qualitative study *	PC	Uncertainty about the diagnosis and lack of adequate information increases the stress on families. Family members prefer fetal cardiologists to clarify and deal with the situation.

CHD: congenital heart disease; PC: pediatric cardiologist; FC: fetal cardiologist; FP: fetal physician. * Application of a questionnaire as a tool for evaluating the effectiveness of family counseling. ** Geneticist, pediatrician, and pediatric surgeon (if necessary).

**Table 3 healthcare-11-02826-t003:** The thematic approach of the 21 included studies.

Thematic Approach	Included Studies
Theme 1: Obstacles to effective family counseling	
Ineffective doctor–patient relationship	1,2,3,4,6,7,8,10,12,13,14,15,21
The information was provided in an inappropriate manner	5,6,8,9,10,12,13,17
Difficulty in standardizing the language	1,3,7,10,11,16,17,18,19,20
Theme 2: Strengths for effective family counseling	
Understanding heart disease	1,3,4,5,9,10,11,12,14,17,19,20,21
The manner in which the diagnosis is communicated	1,2,6,7,12
Good infrastructure	5,8,9,15,17
Theme 3: Opportunities for the healthcare services and the health team	
Focus on humanization	1,2,3,4,5,6,7,8,9,10,11,12,13,14,15,16,17,18,19,20,21
Multidisciplinary support network	6,8,9,10,12,13,17
Comprehension of the disease	5,7,8,10,12,15
Family’s trust in the team	1,2,3,4,7,10,12
Confidence in decision making	1,2,3,4,7,10,12
Theme 4: Challenges for the healthcare services and the health team	
Continuous family counseling	4,6,8,10,12,13,14,15,21
Presence of fetal and pediatric cardiologist	1,3,4,5,9,10,11,12,14,17,19,20,21
Multidisciplinary team	5,6,8,9,10,12,13,17
Audiovisual resources	1,3,7,10,11,16,17,18,19,20

## Data Availability

The data presented in this study are available on request from the corresponding author.

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
