# Peer review of "Family Counseling after the Diagnosis of Congenital Heart Disease in the Fetus: Scoping Review"

_healthcare, 2023, doi:10.3390/healthcare11212826_

Round 1

Reviewer 1 Report

The focus of the review was on family counseling on prenatal diagnosis of congenital heart disease.  21 articles were included in the review. Goal of scoping review: to map the existing scientific evidence and detect gaps for future research (line 52-53) - this is not clear in the abstract. 

The main criticism of this review is the authors need to go more in-depth in their discussion points. They place the burden on the reader to go to the cited study to read more about it. They should instead summarize the main points with enough detail for the reader to understand.  For 21 articles, this review feels very brief.  Below are some of the grammatical errors and lines that I found especially unclear. By no means did I include all of the errors, I suggest a native English speaker provide editing before resubmitting.

Abstract:

1.) First line of abstract is not a complete sentence.

2.) Line 29: "This tool" refers to what?

3.) Overall rationale for the scoping review is not clear.

Introduction:

1.) Line 38: "developing a lot" needs more clarity

2.) Line 41: I assume the incidence is prenatal diagnosis - 1.2% is higher than the reported incidence of CHD globally, this needs to be clarified.

Methods:

1.) Line 65: "bases" should be "basis".

2.) Line 72-73: Rewrite ethics sentence to correct word order.

3.) Line 75-78: Rewrite to be more concise and clear

4.) Line 85-86: Dangling participle, please correct

5.) Line 95: Appears there was a publication deadline for Google Scholar.

6.) Line 96: "not possible" to "not available"

7.) Line 102: "3 reviewer" to "third reviewer"

8.) Elaborate on what they mean by counter-reference and how this and "active search" were carried out.

9.) What do they mean by calibrating an Excel spreadsheet with a randomized sample?

Three themes: 1) obstacles to counseling, 2) strengths for effective counseling, 3) opportunities and challenges to the services and health team.

Results:

1.) Line 119-122: Correct grammatical errors.

2.) Figure 1: Number of studies excluded in the first exclusion step is too high to explain the "registries after the first screening (n=2554)." Number of studies excluded in the second exclusion step is too small to explain the "registries after reading the entire article (n=19)." No explanation of how counter references in 19 articles only yielded 2 more papers.

3.) Figure 2: Countries are not in English. Also need higher-quality images.

4.) Table 2: Title of studies is not relevant to include. I would rather see the methodology of the study, the theme (using the 3 themes the authors defined), number of patients included if available. 

5.) Figure 3: The four circles do not add anything to the image.  I would rather see this in a table or redone differently.  Terms used like "distance," "filter information, "unify the language" need to be better defined.

6.) Figure 4: This needs to be better explained, I don't know what the percentages indicate. Also define "targeted." What qualifies "good infrastructure"?

7.) Figure 5: Words are cut off in the image. The citations really distract from the figure.

8.) Figure 6: I don't understand the link between "Effective Advice" and "Challenges for..." The caption needs to be clearer.

Discussion:

1.) Line 162: "contribution" to "contributor"

2.) What do the authors means by "distance"? Physical? Emotional? Knowledge-based?

3.) Line 164-165: "tool in the CHD's arsenal of greater success" makes no sense.

4.) Describe what "filtering information" means.

5.) Line 174: "researchers" must form close bonds with stakeholders? Where do researchers fit in this, please explain how this relates to family counseling.

6.) Line 185: perhaps the authors can suggest well-vetted internet sources they came across to save the reader from having to read the articles themselves.

7.) Line 188: "elevated" to "presented"

8.) Lines 196: "important others"?  "deliberating the fate"? Rewrite.

9.) Line 202: missing verb. Also, I did not expect pregnancy termination to come up as I consider this in the area of family decision-making - may consider removing this as this is outside the stated scope of the review and may distract readers. Unless they want to include family decision-making, then they need to include other decisions families make prenatally.

10.) Line 210-212: incomplete sentence

11.) Line 212: public policy is also outside the scope of review

12.) Line 212: remove "as needs for effective counseling"

13.) Line 221-223: too long, break it up into 2-3 sentences.

14.) Line 226-228: online and anonymous services are a limiting factor?  This point is entirely unclear.

15.) The rest of Discussion 4.4 is too choppy and unorganized.  This needs a heavy rewrite.

See above.

Author Response

(Reviewer 1)

Abstract:

1.) First line of abstract is not a complete sentence. – It was corrected

2.) Line 29: "This tool" refers to what? – It was corrected

3.) Overall rationale for the scoping review is not clear. – It was corrected

Introduction:

1.) Line 38: "developing a lot" needs more clarity – It was corrected

2.) Line 41: I assume the incidence is prenatal diagnosis - 1.2% is higher than the reported incidence of CHD globally, this needs to be clarified. – It was corrected

Methods:

1.) Line 65: "bases" should be "basis" - it was done

2.) Line 72-73: Rewrite ethics sentence to correct word order - it was done

3.) Line 75-78: Rewrite to be more concise and clear - it was done

4.) Line 85-86: Dangling participle, please correct - it was done

5.) Line 95: Appears there was a publication deadline for Google Scholar. – It was corrected

6.) Line 96: "not possible" to "not available" – It was corrected

7.) Line 102: "3 reviewer" to "third reviewer" – It was corrected

8.) Elaborate on what they mean by counter-reference and how this and "active search" were carried out. – It was corrected

9.) What do they mean by calibrating an Excel spreadsheet with a randomized sample? – we take out this sentence because it was unnecessary

Three themes: 1) obstacles to counseling, 2) strengths for effective counseling, 3) opportunities and challenges to the services and health team. – It was corrected

Results:

1.) Line 119-122: Correct grammatical errors. – It was corrected

2.) Figure 1: Number of studies excluded in the first exclusion step is too high to explain the "registries after the first screening (n=2554)." Number of studies excluded in the second exclusion step is too small to explain the "registries after reading the entire article (n=19)." No explanation of how counter references in 19 articles only yielded 2 more papers. – It was corrected

3.) Figure 2: Countries are not in English. Also need higher-quality images. – we took out figure 2

4.) Table 2: Title of studies is not relevant to include. I would rather see the methodology of the study, the theme (using the 3 themes the authors defined), number of patients included if available. – It was corrected

5.) Figure 3: The four circles do not add anything to the image.  I would rather see this in a table or redone differently.  Terms used like "distance," "filter information, "unify the language" need to be better defined. – we took out figure 3, 4, 5 and 6, and we made the table 3

6.) Figure 4: This needs to be better explained, I don't know what the percentages indicate. Also define "targeted." What qualifies "good infrastructure"? – we took out figure 3, 4, 5 and 6, and we made the table 3. We explained “good infrastructure” in the discussion.

7.) Figure 5: Words are cut off in the image. The citations really distract from the figure. – we took out figure 3, 4, 5 and 6, and we made the table 3.

8.) Figure 6: I don't understand the link between "Effective Advice" and "Challenges for..." The caption needs to be clearer. – we took out figure 3, 4, 5 and 6, and we made the table 3.

Discussion: - we reviewed and rewrote the discussion. The article was revised a native English speaker.

1.) Line 162: "contribution" to "contributor"

2.) What do the authors means by "distance"? Physical? Emotional? Knowledge-based?

3.) Line 164-165: "tool in the CHD's arsenal of greater success" makes no sense.

4.) Describe what "filtering information" means.

5.) Line 174: "researchers" must form close bonds with stakeholders? Where do researchers fit in this, please explain how this relates to family counseling.

6.) Line 185: perhaps the authors can suggest well-vetted internet sources they came across to save the reader from having to read the articles themselves.

7.) Line 188: "elevated" to "presented"

8.) Lines 196: "important others"?  "deliberating the fate"? Rewrite.

9.) Line 202: missing verb. Also, I did not expect pregnancy termination to come up as I consider this in the area of family decision-making - may consider removing this as this is outside the stated scope of the review and may distract readers. Unless they want to include family decision-making, then they need to include other decisions families make prenatally.

10.) Line 210-212: incomplete sentence

11.) Line 212: public policy is also outside the scope of review

12.) Line 212: remove "as needs for effective counseling"

13.) Line 221-223: too long, break it up into 2-3 sentences.

14.) Line 226-228: online and anonymous services are a limiting factor?  This point is entirely unclear.

15.) The rest of Discussion 4.4 is too choppy and unorganized.  This needs a heavy rewrite.

Response: we reviewed and rewrote the discussion. The article was revised a native English speaker.

Reviewer 2 Report

The whole article needs to be revised by a professional in regard to use of language, poor English grammar and choice of words makes it hard to understand what the authors had in mind. Also the article is in need of serious use editing and usage of punctuation.

Abstract:

The first line states the aim of the study without saying it is the aim of the study.

Line 16- try using were searched rather then used

Lines 19-24- why do you use( ;)  and not ( ,) between mentioned inclusion criteria, the statement in line 22 beginning with inclusion should be put in brackets

Line 24- it is good to mention that out of bigger search you chose 21 articles

Line 25- the usage of English is poor it indicates that your research not researches you chose to evaluate were done in a specific manner

The results of your study are not clearly stated in your abstract, they rather feel like loos observations. And out of the blue in last sentence you state same suggestions without clearly stating what let you to that specific conclusion.

Introduction:

Lines 42-43- use of language

Line 44- poor choice of word- didactic

Line 49- did you mean: that parents have many doubts while receiving information about CHD??

Line 51- decide whether you choose to use either acronym CHD or not but try to be consistent

Line 52- rather then writing we decided try to define the aim of the study

Line 53- the objective of this study rather then objective this study

Line 55- what do you mean by demonstrating gaps? What gaps? Define that.

Materials and Methods:

Lines 64-67 seem to be irrelevant once you explain in chapter above that you chose to use PRISMA-ScR

Lines-72-73- correct grammar

2.1. Do I understand correct that you chose only one question quite a broad one while conducting your literature review?

Table 1 is to large (takes up two pages) needs editing: widen the search column while narrowing the rest of the columns

2.3. Line 94-95 again grammar an language must be corrected

Line 98- grammar

2.4. Line 108- what do you mean by important results of the study?? Please explain.

Results

Figure 1 is not understandable and inconsistent:

-          Out of 2554 studies you name reasons for exclusion that ad up to 2594 exclusions, nevertheless you register in the next step 65 studies out of which you again exclude 32 (stating the reasons) and come with a number of 19 and not 33??? Why

-          At the and you get 21 included studies out of 19??? It is not clear why?

Figure 2 is not clear to read and not necessary in this review, you mentioned in the text that the only studies you found regarded those done in developed countries.

Figures 3-6 need reediting they are not readable, fond of the descriptions is to small, the choice of figures is also not consistent and poor.

Discussion

Line 155- your statement says lack of research did you mean that the research is scares???Or did you mean to underline that there are some parts of the word where such studies are not done??

Then you have sections from 4.1-4.5 try to unify the titles in the first one you state “in case of CHD”  but you do not do that for the other sections

4.1.

Line 166- language choice, by decision makers did you mean parents? Health providers? , word stunned is a colloquialism,

4.2

You repeat your argument form section 4.1. in lines 173-176

Line 177- language/grammar

Line 179- grammar

Line 180- grammar

 The whole chapter is poorly written in regard to grammar and language thus it is hard to understand

Line 194- repetition

Line 196- grammar

4.3

Line 199- poor choice of word building

Line 202- grammar

Line 204- poor choice of word defenders, grammar

Line 210- a sentence without subject and predicate

Line 216-218- poor language

4.4

Line 221- grammar, I do not understand what you mean by this sentence

Line 229-230- grammar

Lines 237-240- not relevant to the articles purpose

Line 245- grammar

Lines 250-251-grammar

Lines 253-256- are not clear, what did you have in mind?

4.5

You mention only one limitation, please state more like few questions you analyzed and unsewered, lack of unity of the analyzed studies …

Conclusion

Line 271- you said in section 4.5 you did not asses the quality of studies analized but than you state they had a low level of evidence, so which is it?

Line -275- grammar/ language

Line 276- I would be careful  giving the statement that your work paves the main road, please rephrase

Needs extensive language and grammar correction

Author Response

(Reviewer 2)

Abstract: – It was corrected

The first line states the aim of the study without saying it is the aim of the study.

Line 16- try using were searched rather then used

Lines 19-24- why do you use ( ;)  and not ( ,) between mentioned inclusion criteria, the statement in line 22 beginning with inclusion should be put in brackets

Line 24- it is good to mention that out of bigger search you chose 21 articles

Line 25- the usage of English is poor it indicates that your research not researches you chose to evaluate were done in a specific manner

The results of your study are not clearly stated in your abstract, they rather feel like loos observations. And out of the blue in last sentence you state same suggestions without clearly stating what let you to that specific conclusion.

Introduction: – It was corrected

Lines 42-43- use of language

Line 44- poor choice of word- didactic

Line 49- did you mean: that parents have many doubts while receiving information about CHD??

Line 51- decide whether you choose to use either acronym CHD or not but try to be consistent

Line 52- rather then writing we decided try to define the aim of the study

Line 53- the objective of this study rather then objective this study

Line 55- what do you mean by demonstrating gaps? What gaps? Define that.

Materials and Methods:

Lines 64-67 seem to be irrelevant once you explain in chapter above that you chose to use PRISMA-ScR – we excluded the paragraph

Lines-72-73- correct grammar – It was corrected

2.1. Do I understand correct that you chose only one question quite a broad one while conducting your literature review? - Yes. Since this scoping review, we think that one broad question could fit to this study design

Table 1 is to large (takes up two pages) needs editing: widen the search column while narrowing the rest of the columns – I believe that this edition will be provided by the editorial team

2.3. Line 94-95 again grammar an language must be corrected – It was corrected

Line 98- grammar – It was corrected

2.4. Line 108- what do you mean by important results of the study?? Please explain. – It was corrected

Results

Figure 1 is not understandable and inconsistent: – It was corrected

-          Out of 2554 studies you name reasons for exclusion that ad up to 2594 exclusions, nevertheless you register in the next step 65 studies out of which you again exclude 32 (stating the reasons) and come with a number of 19 and not 33??? Why – It was corrected

-          At the and you get 21 included studies out of 19??? It is not clear why? – It was corrected

Figure 2 is not clear to read and not necessary in this review, you mentioned in the text that the only studies you found regarded those done in developed countries. – we took out figure 2

Figures 3-6 need reediting they are not readable, fond of the descriptions is to small, the choice of figures is also not consistent and poor. – we took out figure 3, 4, 5 and 6, and we made the table 3.

Discussion - we redo the whole discussion

Line 155- your statement says lack of research did you mean that the research is scares???Or did you mean to underline that there are some parts of the word where such studies are not done??

Then you have sections from 4.1-4.5 try to unify the titles in the first one you state “in case of CHD” but you do not do that for the other sections

4.1.

Line 166- language choice, by decision makers did you mean parents? Health providers? , word stunned is a colloquialism,

4.2

You repeat your argument form section 4.1. in lines 173-176

Line 177- language/grammar

Line 179- grammar

Line 180- grammar

 The whole chapter is poorly written in regard to grammar and language thus it is hard to understand

Line 194- repetition

Line 196- grammar

4.3

Line 199- poor choice of word building

Line 202- grammar

Line 204- poor choice of word defenders, grammar

Line 210- a sentence without subject and predicate

Line 216-218- poor language

4.4

Line 221- grammar, I do not understand what you mean by this sentence

Line 229-230- grammar

Lines 237-240- not relevant to the articles purpose

Line 245- grammar

Lines 250-251-grammar

Lines 253-256- are not clear, what did you have in mind?

4.5

You mention only one limitation, please state more like few questions you analyzed and unsewered, lack of unity of the analyzed studies …

Response: we redo the whole discussion

Conclusion - we rewrote the whole conclusion

Line 271- you said in section 4.5 you did not asses the quality of studies analized but than you state they had a low level of evidence, so which is it?

Line -275- grammar/ language

Line 276- I would be careful giving the statement that your work paves the main road, please rephrase

Response: we rewrote the whole conclusion

Reviewer 3 Report

I have only one comment:

Please add a section in the discussion part with the quality of the selected articles.

There are some grammatical and syntax errors in the manuscript.

Author Response

(Reviewer 3)

Please add a section in the discussion part with the quality of the selected articles. – we add the studies methods in Table 3 and add the limitation of this study design in provide a systematic approach to evaluate the different quality of the studies.

Round 2

Reviewer 2 Report

Thenk you for making suggested corrections, now the article is suited to be admited to publication.

Minor English corrections to be made by editorial team